# Feasibility trial of Darwin OncoTreat and OncoTarget precision medicine testing to improve outcomes for patients with limited metastatic disease that failed first-line systemic therapy

Rachel Radigan[1,2]*, Ashish Sangal[3], Dong Zhang[2], Jonathan Weber[4], Megha Schmalzle[1,2], Alexa Finkelstein[2], Vaibhav Duggal[2], Johnny Kao[1,2,3]

1 Department of Radiation Oncology, Good Samaritan University Hospital, West Islip, New York, United States of America, 2 Center for Cancer Research, New York Institute of Technology College of Osteopathic Medicine, Greenvale, New York, United States of America, 3 Cancer Institute at Good Samaritan University Hospital, West Islip, New York, United States of America, 4 The DeMatteis Center for Cardiac Research and Education, Greenvale, New York, United States of America

* rradigan@nyit.edu

## Abstract

Despite significant progress in solid tumor oncology, including widespread genomic testing, metastatic cancer remains largely incurable and results in approximately 90% of cancer deaths. In the context of systems biology, RNA transcriptome-based (RNA-seq) testing is utilized to identify master regulator proteins that are putative drivers of tumor progression. After extensive preclinical testing and validation, Darwin OncoTarget and OncoTreat has been developed as a commercially available next generation precision oncology test with preliminary evidence of efficacy in treatment-refractory advanced cancers. We designed this pilot trial to assess the feasibility of integrating Darwin OncoTarget and OncoTreat testing in patients with oligometastases receiving comprehensive involved site radiotherapy. Eligible patients are adults with solid tumor oligometastases with up to 10 discrete tumors amenable to radiation therapy following prior first-line systemic therapy. Tumor biopsy is required to allow for precision medicine testing to supplement standard clinical management. Formalin fixed paraffin embedded tissue with >50% tumor will be sent to the Laboratory of Personalized Genomic Medicine at Columbia University Medical Center for Darwin OncoTarget and OncoTreat testing. Patients will either continue standard of care systemic therapy or proceed with an alternative FDA approved treatment informed by Darwin testing. This trial evaluates the feasibility and utility of integrating novel precision oncology testing in a community hospital setting. This study will utilize precision oncology testing in the population of induced, recurrent or persistent oligometastases that currently have limited or largely ineffective systemic treatment options. This trial represents an early attempt at integrating next generation precision medicine testing and systems biology in the context of radiation therapy.

**Data availability statement:** No datasets were generated or analyzed during the current study. All relevant data from this study will be made available upon study completion and approval from the IRB upon request.

**Funding:** The author(s) received no specific funding for this work.

**Competing interests:** The authors have declared that no competing interests exist.

**Abbreviations:** FDA, Food and Drug Administration; SHIVA trial, Molecularly targeted therapy based on tumor molecular profiling versus conventional therapy for advanced cancer; NCI-MPACT trial, Molecular Profiling-Based Assignment of Cancer Therapy; RNA, Ribonucleic Acid; CLIA, Clinical Laboratory Improvement Amendments of 1988; ESTRO EORTC, European Society for Radiotherapy and Oncology and the European Organisation for Research and Treatment of Cancer; SABR-COMET-10 trial, Stereotactic Ablative Radiotherapy for the Comprehensive Treatment of Oligometastatic Disease; NEAT, NEAT model evaluates the number of active tumors ("N"), Eastern Cooperative Oncology Group performance status ("E"), albumin ("A") and primary tumor site ("T"); OHSU, Oregon Health & Science University; DNA, Deoxyribonucleic Acid; CTCAE, Common Terminology Criteria for Adverse Events (CTCAE); IRB, Institutional Review Board.

## Background

There is great enthusiasm for advances in drug development targeting distant metastases from solid tumors in the mainstream media [1]. Despite significant progress, metastatic cancer remains largely incurable and results in approximately 90% of cancer deaths [2]. Following treatment with either immunotherapy or molecularly targeted systemic therapies alone, responses are uncommon benefiting approximately 15% of all cancer patients [3,4]. Published evidence dating to the late 2000's established the long-term curative potential of radiation therapy to all areas of known disease for patients with oligometastases, historically defined as 5 or fewer clinically detectable metastatic lesions [5]. Two randomized trials demonstrated improved progression-free survival and overall survival when comprehensive local consolidative therapy is added to systemic therapy alone for patients with oligometastases from non-small cell lung cancer or mixed primary tumors [6,7]. A more recent randomized trial specifically demonstrated a large benefit for irradiating oligoprogressive tumors from non-small cell lung cancer [8]. By contrast, adding stereotactic radiation to some but not all distant metastases failed to improve outcomes compared to immunotherapy alone [9,10].

Good Samaritan University Hospital has been a leader in utilizing comprehensive involved site radiotherapy for oligometastases. Among 130 patients with oligometastases treated from 2014 to 2021, the median overall survival is 36 months with a 4-year overall survival of 41% [11]. The 4-year progression-free survival was 23% with a median progression-free survival of 13 months. The 4-year progression-free survival was 23% with only 1 patient developing isolated local failure. In terms of patterns of failure, the 4-year local control was 86% and the 4-year distant control was 29%. Based on this analysis, preventing additional distant metastases is a prerequisite for more effective treatment for oligometastases.

While drug development continues, new agents are generally tested as single agents with the ultimate goal of obtaining FDA approval. With few exceptions, novel cancer drugs are not tested in combination with radiotherapy [12]. Therefore, while comprehensive radiation for oligometastases is surprisingly effective, it has been difficult to further improve progression-free survival over the past decade and a half [13].

Following the Cancer Genome Project, there was great excitement about the potential of genomic testing to improve systemic therapy for patients with metastatic cancer [14]). Two high profile randomized trials, the SHIVA and NCI-MPACT trials, were both negative [15,16]. The SHIVA trial achieved a median progression-free survival of 2 months despite genomic testing and the NCI-MPACT trial had a 2% objective response rate [15,16]. As memorably stated by Dr. Ian Tannock, "the concept of personalized medicine is so appealing that seemingly only curmudgeons could criticize it" [14]. Despite lack of efficacy, genomic testing is a current standard of care for many metastatic malignancies using commercial platforms including Foundation One and CARIS with an estimated 5% response rate with median duration of response of 29.5 months [4,17]. Obviously, genomic testing alone is inadequate for the vast majority of patients with solid tumor metastases.

With advances in science and computational power, we are now in the era of systems biology [18]. One proposed precision medicine approach is using RNA transcriptome-based (RNA-seq) testing to identify master regulator proteins that are putative drivers of tumor progression [19]. OncoTarget and OncoTreat predict potential drugs with early markers of efficacy in early human clinical trials. Specifically, OncoTarget identifies high-affinity inhibitors of master regulator proteins, while OncoTreat identifies tumor-checkpoint module inhibitors that modulate the transcriptional activity of hyper-connected master regulators [19]. These tests are now commercially available with CLIA approval through Columbia Presbyterian Medical Center. Comparative studies evaluating the clinical efficacy of OncoTarget versus OncoTreat are currently limited, although a past trial at Columbia University studied the efficacy of OncoTarget and OncoTreat in 7 patients. The endpoint was 30-day disease control based on single agent activity with OncoTarget (68%) and OncoTreat (91%) both highly significant with the predicted drugs outperforming control of selected antineoplastic drugs [19]. We propose supplementing routine clinical care with OncoTarget and OncoTreat precision medicine testing with the goal of improving outcomes. Among ESTRO EORTC oligometastasis subgroups, patients with oligoprogression and oligorecurrence following prior standard of care systemic therapy are the most logical candidates for precision medicine testing [20]. Moreover, data from Good Samaritan University Hospital demonstrates that patients with induced or recurrent oligometastases have a median progression-free survival of 9 months vs. 14 months for de novo oligometastases (p = 0.04 for induced vs. de novo oligometastases) [21]. We propose utilizing precision oncology testing in the population of induced or recurrent oligometastases that currently have limited or largely ineffective systemic treatment options. The feasibility of Precision Medicine testing in a community oncology setting relying predominantly on reimbursement for clinical services is an important endpoint of this study.

## Methods and design

### Aims and objectives

The aims of this study include two feasibility endpoints. The first is to evaluate the feasibility of OncoTarget and OncoTreat testing using a basket design of patients with oligoprogression or oligopersistence across various solid tumor histology. We define feasbility as four different endpoints, which include: 1) ability to have the OncoTarget and OncoTreat test performed based on tumor type and pathology with sufficient cellularity/DNA/RNA; 2) willingness of the treating medical oncologist to utilize FDA approved but off label medication; 3) ability to procure FDA approved on-label and/or off-label agents recommended by OncoTarget and OncoTreat testing through insurance or compassionate use in the setting of extensive preauthorization requirements prevalent in community oncology practice; and 4) identify any unknown barriers. The second feasbility endpoint is to evaluate the proportion of cases where OncoTarget and OncoTreat testing, compared to standard of care genomic testing including Foundation One or CARIS, results in a change in choice of systemic treatment that can be implemented in a community hospital setting. Although neither a primary or secondary endpoint, we will assess progression-free survival defined as the time between enrollment and tumor progression or death for patients with oligometastases treated with comprehensive involved site radiotherapy compared to historical control patients treated without OncoTarget and OncoTreat testing. Finally, we will compare the progression-free survival of radiation combined with precision oncology recommended treatment to the progression-free survival of the previous line of treatment.

### Participant recruitment

A total of 20 patients with recurrent or residual disease following first-line or later systemic therapy will be treated as part of a feasibility trial. Eligible patients are patients aged ≥18 years with metastatic solid tumor who are candidates for comprehensive involved site radiotherapy to all areas of disease seen on whole body imaging. Patients with brain metastases are eligible provided they qualify as oligometastases (up to 10 lesions based on the SABR-COMET-10 trial) and are treatable with radiotherapy to all areas of visible disease [22].

Patients with an estimated median survival of less than 6 months using the published and validated NEAT (number of tumors, ECOG performance status, albumin and tumor type) methodology are ineligible [23,24]. The NEAT model combines 4 independent prognostic factors to create a composite score (Table 1). Based on updated data, for patients with 1–5 active tumors, patients with a NEAT score of 0 to 3.5 have a median survival > 1 year. For patients with ≥6 active tumors, only patients with a NEAT score of 0–2 have a median survival > 6 months [25]. Eligible patients have adequate bone marrow (absolute neutrophil ≥1000, hemoglobin ≥ 9 g/dl, platelets ≥100,000), kidney (creatinine ≤1.5 times upper limit of normal) and liver (bilirubin ≤1.5 times upper limit of normal). Patients who are pregnant and/or breast feeding or those with severe uncontrolled cardiac, pulmonary, infectious or organic brain disease are ineligible.

## Study design and oversight

This is a feasibility trial, NCT#06842030, investigating OncoTarget and OncoTreat testing in patients eligible for comprehensive involved site radiotherapy for up to 10 metastases [26]. Prior to precision medicine testing, the following baseline clinical characteristics will be collected using the patient's electronic medical record: age, sex, ECOG performance status, serum albumin, synchronous or metachronous oligometastatic disease, location of primary and metastatic tumor(s), number of active tumors, type of systemic therapy and prior radiotherapy type. All patients will undergo mandatory tumor biopsy based on their standard-of-care treatment plan. The biopsy will be performed by the patient's typical clinical care team, and then formalin-fixed paraffin-embedded tissue with >50% tumor will be sent to the Laboratory of Personalized Genomic Medicine at Columbia University Medical Center. Testing will not be able to be performed on those whose tumor tissue biopsies contain less than 50% of tumor; they will continue standard-of-care therapy. The cost of testing is $1600 per specimen and the cost will be generously covered by the Good Samaritan Hospital Foundation. The turnaround time of Darwin Health testing is 21 days. During that waiting period, patients will proceed with radiation simulation accomplished by computed tomography (CT) imaging, treatment planning and delivery of radiation therapy to all areas of visible disease. During or after radiotherapy, patients can receive standard of care systemic therapy at the discretion of the treating oncologist. Patients will continue to be treated by the medical oncologist to avoid any gaps in care while awaiting Darwin OncoTarget and OncoTreat test results as well as acquisition of recommended agent. Once the OncoTarget and OncoTreat recommendations are available, the treating oncologist can administer standard of care systemic therapy or proceed with treatment recommended by precision medicine testing. If there are delays in securing access to OncoTarget and OncoTreat recommended systemic therapy beyond 2 weeks after receiving results of the recommended therapy, the treating oncologist can initiate standard of care systemic therapy. Overview of the enrollment, interventions and assessment process is depicted in Fig 1. The overall treatment schema is displayed in Fig 2.

In terms of systemic treatment and follow-up, patients will undergo standard of care history, physical examination, toxicity monitoring and tumor imaging. The testing to be performed and the study will be reviewed with patient and/or family members.

**Table 1. Derivation of the validated NEAT prognostic model.**

| Variable | Number of active tumors | ECOG performance status | Serum albumin (g/dl) | Primary tumor site |
|---|---|---|---|---|
| 0 points | 1 to 5 | 0 to 1 | ≥3.4 | Breast, prostate or kidney |
| 0.5 points | | | 2.4 to 3.3 | |
| 1 point | ≥6 | 2 | <2.4 | Other tumor sites |
| 2 points | | 3–4 | | |

The estimated end of the recruitment period for this study will be when n of 20 is reached, or after the interim analysis of n of 5 dependent on analysis results.

| TIMEPOINT | Enrollment | Allocation 0 | Post-Allocation 0 | + 3-4 weeks | Every 3 months (Year1) | Every 6 months (+Year1) | Close-Out Expiration |
|---|---|---|---|---|---|---|---|
| **ENROLLMENT:** | | | | | | | |
| Eligibility Screen | X | | | | | | |
| Informed Consent | X | | | | | | |
| Tumor Biopsy | | X | | | | | |
| **INTERVENTION:** | | | | | | | |
| Darwin OncoTarget & OncoTreat Testing | | | X | | | | |
| Treatment Recommendation | | | | X | | | |
| **ASSESSMENTS:** | | | | | | | |
| Medical Oncologist Treatment Decision | | | | X | | | |
| Ability to Acquire Recommended Treatment (Insurance Coverage) | | | | X | X | | |
| Ability to Acquire Recommended Treatment (Compassionate Use) | | | | | X | X | X |
| Various Solid Tumor Histology Applicability | | | | X | X | X | X |
| Progression-Free Survival | | X | X | X | X | X | X |
| Standard of Care Follow Up | | X | X | X | X | X | X |

**Fig 1. SPIRIT figure timeline.** Timeline demonstrates schedule of enrollment, interventions and assessments. Ongoing oversight for this trial is available on ClinicalTrials.gov (NCT06842030).

Baseline and follow-up circulating tumor DNA testing (using commercially available vendors) is optional but encouraged as part of the standard of care. Patients will be followed as per standard of care until death. Typically, patients diagnosed and treated by radiation oncology and/or medical oncology are seen at least every 3 months for the first year and every 6 months thereafter. The overall study timeline is displayed in Fig 3. All oncology patients who are hospitalized at Good Samaritan University Hospital are jointly reviewed by medical oncology and radiation oncology at a daily huddle Monday through Friday.

Toxicity will be assessed using the CTCAE scoring system of grade 1–5. Toxicities will be attributed to radiotherapy (acute vs. late) or systemic therapy. Any serious adverse event defined as CTCAE grade 4 or 5 toxicities will be reported to the Good Samaritan University Hospital IRB. Any patient who experiences an adverse event may have the putative offending therapy discontinued or dose adjusted, as determined by the treating oncologist.

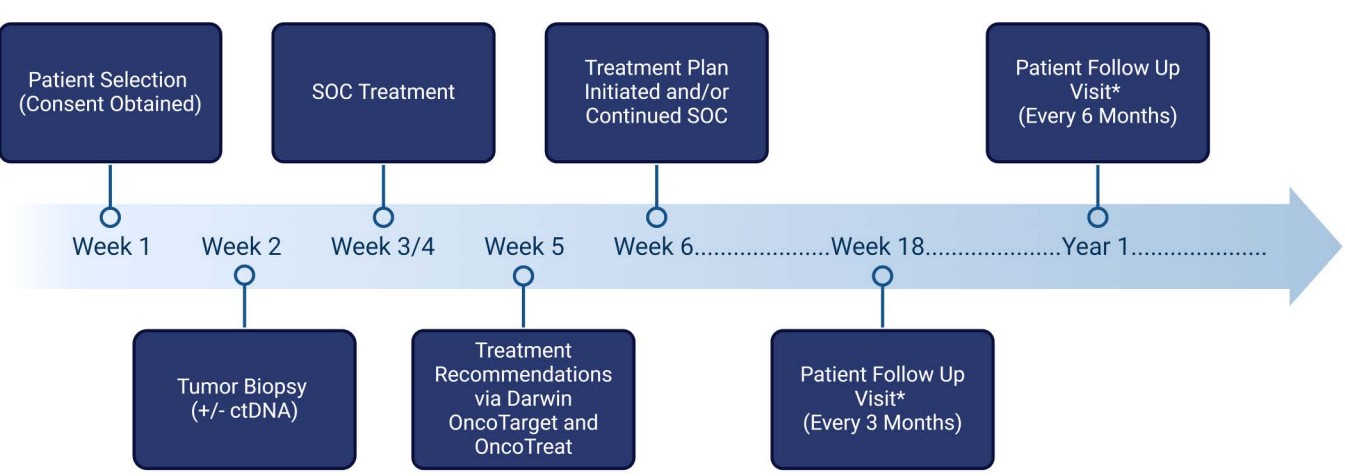

**Fig 2. Protocol overview.** Process and design scheme of study design. Created with BioRender.com.

SOC: Standard of Care
*Patient follow up visits consist of history, physical examination, toxicity monitoring, and tumor imaging at each visit.

**Fig 3. Study timeline.** Created with BioRender.com.

This study received ethics approval by the Good Samaritan University Hospital IRB on March 13, 2024 (Protocol #24-002). It is registered on ClinicalTrials.gov. This trial will follow the ethical guidelines set forth by the Declaration of Helsinki. Patients must sign informed consent, which will be facilitated by study personnel. Patient information will remain confidential with only essential study personnel having access to study data stored in electronic files on a password-protected computer managed by Catholic Health Information Technology. This protocol was written by the authors; the funders did not contribute to the design, performance, or reporting. Relevantprotocol modifications and results will be communicated through the clinical trial registry. Clinicaltrials.gov will be updated as the study progresses and finalizes, and results will be used for publication of manuscript. Fig 1 demonstrates schedule of enrollment, interventions and assessments from enrollment to study completion.

## Treatment recommendations and outcomes

An important primary outcome measure of this trial is the proportion of participants whose tumor biopsy successfully undergoes OncoTarget and OncoTreat testing based on sufficient cellularity/DNA/RNA, defined as >50% tumor. When the treating medical oncologist receives the test results, they will participate in multidisciplinary discussion and decide to treat with usual second-line or later systemic therapy or instead utilize precision medicine recommendations. Therefore, off-label use of FDA approved agents for patients with historically incurable metastatic cancer will occur outside the auspices of this clinical trial focused on Precision Medicine testing. Progression-free survival, defined as the time between study enrollment and tumor progression or death, will be recorded and compared to historical control patients who were treated without OncoTarget and OncoTreat testing.

For research and reporting purposes, we will compare OncoTarget and OncoTreat recommendations vs. Foundation One and Caris genomic recommendations [27]. The proportion of cases in which each precision medicine test influenced management plans will be evaluated. WhileOncoTarget and OncoTreat are not yet used at other community sites, multiple site-specific tumor boards, molecular tumor board and consultation with Roswell Park Cancer Institute physicians are available to assist the treating medical oncologist.

A final feasibility outcome measure is the proportion of precision medicine recommendations that are actionable in the setting of extensive preauthorization requirements prevalent in community oncology practice in the Unites States of America. From the experience of Dr. Gordon Mills at Oregon Health and Science University Knight Cancer Institute, insurance coverage can be obtained for approximately 60% of patients through appealing insurance denials (personal communication). When insurance coverage is not possible despite appeals, every effort will be made to obtain compassionate use through the drug manufacturer. Data will be collected on the process of accessing precision medicine recommended drugs to determine feasibility of Darwin OncoTarget and OncoTreat testing.

## Statistical analysis

A prior feasibility trial of OncoTreat and OncoTarget reported by Columbia University only consisted of 7 patients [19]. Since patients with oligometastases treated with radiotherapy are expected to have a longer progression-free survival, we expanded the sample size to 20 patients based on the feasibility objectives of the study. This includes evaluating the ability to perform OncoTarget and OncoTreat testing, integrating treatment recommendations, and assessing insurance coverage for drug use. An interim analysis will be performed after enrolling5 patients to ensure feasibility with future study decisions made by the principal investigator.

Analysis of feasibility is mainly descriptive. To evaluate proportion of cases where treatment choice changes, a z-test will be used to predict statistically significant differences from standard of care ($\alpha = 0.05$). Subgroup analyses based on the number of active tumors (1–5 vs. 6–10) and primary tumor type will be considered to assess for variability in feasibility endpoints. Progression-free survival and overall survival will be compared to historical controls using the published NEAT algorithm with 95% confidence intervals for the hazard's ratios [23]. To compare survival between patients treated based

on OncoTarget and OncoTreat recommendations and historical controls, a log-rank test will be used. A von Hoff ratio of greater than 1.3 is accepted as a measure of exceptional response.

Any participants who may deviate from the protocol (e.g., discontinuation of the recommended treatment due to adverse events or insurance barriers) will be documented and analyzed descriptively to assess their impact on feasibility outcomes. To address potential missing data for key outcomes, multiple imputation methods will be applied by estimating missing values based on observed data to reduce bias and maintain validity. Sensitivity analyses will be conducted to compare results with and without imputation to ensure robustness.

## Discussion

Practical and operational issues involved in performing this study include ability to perform Darwin precision oncology testing on most cancer types and the ability obtain off label use of the recommended OncoTarget or OncoTreat drug [28].

The precision medicine recommendations may include off-label use of FDA approved agents. As such obtaining insurance coverage of these costly agents may be difficult or impossible to obtain. Every attempt will be made to appeal initial insurance denials in collaboration with the treating medical oncologist. If coverage continues to be unattainable, we will apply for compassionate use directly through the drug manufacturer. A final option is using the Mark Cuban Cost Plus Drug platform although availability of FDA approved generic targeted therapies is currently extremely limited [29]. Despite these efforts, it is possible the Darwin OncoTarget and OncoTreat recommended drugs are still not able to be affordably secured for the patient. While this is an issue that may be encountered, this is part of the outcome measures for this feasibility study.

Ovarian cancer, cholangiocarcinoma and stomach cancer are considered suboptimal candidates for Darwin testing due to mixed cellular subpopulations resulting in potentially inaccurate predictions due to the nature of bulk tumor rather than single cell precision oncology testing (personal communication with Columbia Presbyterian Medical Center).

The projected practical and operational issues of this study are part of the outcome feasibility measures to determine if personalized medicine through Darwin OncoTarget and OncoTreat can be useful across an array of cancer types in a community hospital setting. By conducting this precision medicine test on a maximum of 20 patients with diverse cancer characteristics, this trial is comparable to an n-of-1 design in which we can assess each patient's outcome after initiating an individualized treatment plan based on their unique tumor profile [30]. With potential promising results of this feasibility n-of-1 trial and identification of potential barriers as part of the learning curve, our ultimate goal is a randomized controlled trial of Darwin precision medicine testing designed to demonstrate superior survival for patients with distant metastases. The integration of genomic analyses into a randomized controlled trial design to personalize treatment strategies and potentially enhance patient outcomes is becoming more widely adopted [31]. Despite significant anticipated challenges, this trial represents an early attempt at applying the potential next generation precision oncology testing informed by systems biology to reducing the risk of distant progression in patients with treated oligometastases.

## Supporting information

**S1 Checklist. SPIRIT checklist – Darwin protocol PLOS One.**
(PDF)

## Author contributions

**Conceptualization:** Rachel Radigan, Ashish Sangal, Dong Zhang, Johnny Kao.

**Formal analysis:** Jonathan Weber, Megha Schmalzle.

**Investigation:** Ashish Sangal, Johnny Kao.

**Methodology:** Rachel Radigan, Ashish Sangal, Dong Zhang, Jonathan Weber, Johnny Kao.

Project administration: Rachel Radigan, Megha Schmalzle, Johnny Kao.

Resources: Rachel Radigan, Megha Schmalzle, Johnny Kao.

Supervision: Johnny Kao.

Validation: Jonathan Weber, Johnny Kao.

Visualization: Rachel Radigan.

Writing – original draft: Rachel Radigan, Jonathan Weber, Alexa Finkelstein, Vaibhav Duggal, Johnny Kao.

Writing – review & editing: Rachel Radigan, Ashish Sangal, Dong Zhang, Megha Schmalzle, Alexa Finkelstein, Vaibhav Duggal, Johnny Kao.

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
