## [Decision Letter · Decision Letter 0]

8 Jan 2025

Dear Dr. Radigan,

We look forward to receiving your revised manuscript.

Kind regards,

Andres Mauricio Acevedo-Melo, M.D.

Academic Editor

PLOS ONE

Journal Requirements:

2. Please ensure you have included the registration number for the clinical trial referenced in the manuscript.

4. Your abstract cannot contain citations. Please only include citations in the body text of the manuscript, and ensure that they remain in ascending numerical order on first mention.

Reviewers' comments:

Reviewer's Responses to Questions

**Comments to the Author**

1. Does the manuscript provide a valid rationale for the proposed study, with clearly identified and justified research questions?

Reviewer #1: Yes

Reviewer #2: Yes

2. Is the protocol technically sound and planned in a manner that will lead to a meaningful outcome and allow testing the stated hypotheses?

Reviewer #1: Partly

Reviewer #2: Yes

3. Is the methodology feasible and described in sufficient detail to allow the work to be replicable?

Reviewer #1: Yes

Reviewer #2: Yes

4. Have the authors described where all data underlying the findings will be made available when the study is complete?

Reviewer #1: No

Reviewer #2: Yes

5. Is the manuscript presented in an intelligible fashion and written in standard English?

Reviewer #1: Yes

Reviewer #2: Yes

You may also provide optional suggestions and comments to authors that they might find helpful in planning their study.

Reviewer #1: This manuscript sets out the protocol for evaluating the feasibility of a novel genomic RNA test to aid precision medicine decisions in treating metastatic cancer. The authors describe the rationale well and designed a trial to assess the possibility of integrating the precision medicine test into the standard care pathway. I appreciate that this is an early stage trial and including patients with metastatic cancer is challenging. Please see my suggestions for improving the manuscript. I hope you find this helpful.

1. Please can you explain why there is no ongoing oversight for this trial. The SPIRIT checklist is NA for sponsor and committees for example. If these are not required for this trial design, please include the rationale for this on the SPIRIT checklist, or if proxies for this oversight are available (i.e IRB or PI responsibility).

2. Can you give some more justification or rationale for the sample size target of 5-20 patients. Is this in relation to how many patients this institution cares for, for example? Can you reasonably expect meaningful statistical results from 5 patients?

3. Can you include in the supplementary materials the baseline patient and clinical characteristics and endpoint measures, for example, in a simple proforma – presumably you are collecting this information somehow? And when these assessments are made, if at different times during the trial.

a. Also, in Fig1, I am not clear what Q is in Q3 months and Q6 months.

4. Can you make the following methods clearer…

a. (line 161 & Fig1) What happens to those patients with samples <50% tumor?

b. (lines 159-165) can you make it clearer who is doing the biopsy and that Columbia University Medical Center is doing the Darwin testing. I think this is clearer in Fig 3, than it is in the text.

c. Are you going to report how many patients were ineligible for patient selection overall and why (e.g. no consent, NEAT score)? This would also be useful to know for patient acceptability – e.g. out of how many patients >18 with up to 10 lesions and classify as oligometastases, were included in the trial.

5. Would the long term goal be for an RCT to evaluate the efficacy of precision medicine using the Darwin tests for metastatic cancer survival? If so, would it be useful to include the acceptability of randomisation to precision medicine, in theory, for both patients and clinicians?

Reviewer #2: Very Minor Thought

This appears to be a well-designed feasibility trial with the feasibility objectives to be correctly summarised without formal statistical analysis. As the authors will no doubt concede, statistical significance is unlikely when the suggested comparisons of progression free survival and overall survival are made. However, 95%CI intervals for the HRs estimated would be informative.

**Do you want your identity to be public for this peer review?** For information about this choice, including consent withdrawal, please see our Privacy Policy

Reviewer #1: No

Reviewer #2: No

---

## [Author Response · Author response to Decision Letter 1]

27 Apr 2025

REVIEWER #1: This manuscript sets out the protocol for evaluating the feasibility of a novel genomic RNA test to aid precision medicine decisions in treating metastatic cancer. The authors describe the rationale well and designed a trial to assess the possibility of integrating the precision medicine test into the standard care pathway. I appreciate that this is an early stage trial and including patients with metastatic cancer is challenging. Please see my suggestions for improving the manuscript. I hope you find this helpful.

1. Please can you explain why there is no ongoing oversight for this trial. The SPIRIT checklist is NA for sponsor and committees for example. If these are not required for this trial design, please include the rationale for this on the SPIRIT checklist, or if proxies for this oversight are available (i.e IRB or PI responsibility).

Response: Good Samaritan University Hospital IRB is overseeing this trial. Additionally, the trial has been PRS-approved and is currently available to view on ClinicalTrials.gov (NCT06842030). This has been mentioned in the SPIRIT checklist.

2. Can you give some more justification or rationale for the sample size target of 5-20 patients. Is this in relation to how many patients this institution cares for, for example? Can you reasonably expect meaningful statistical results from 5 patients?

Response: Due to the nature of our patient population and our feasibility trial design, 20 patients were set to determine feasibility outcomes of the Darwin OncoTarget and OncoTreat test. An interim analysis of 5 is to ensure continued benefice to patients. Our numbers are also based on a prior feasibility study of only 7 patients and we hope to expand this to gather further information about practicality within a community hospital setting.

3. Can you include in the supplementary materials the baseline patient and clinical characteristics and endpoint measures, for example, in a simple proforma – presumably you are collecting this information somehow? And when these assessments are made, if at different times during the trial.

Response: Baseline clinical characteristics will be collected prior to biopsy and precision medicine testing; the specific data elements are detailed in Study Design. Post-test endpoint measures include feasibility of completing the test, changing treatment regimen, procuring prescribed drugs through insurance, and measuring progression-free survival. These endpoints along with toxicities are explained in the Study Design and OncoTarget and OncoTreat Treatment Recommendations.

a. Also, in Fig1, I am not clear what Q is in Q3 months and Q6 months.

Response: Thank you for pointing this out. Abbreviated Q for “every”. Updated Fig 1 to avoid abbreviation and make state “every 3 months” and “every 6 months”.

4. Can you make the following methods clearer…

a. (line 161 & Fig1) What happens to those patients with samples <50% tumor?

Response: Thank you for your comment. This is one of our feasibility outcomes: 1) ability to have the OncoTarget and OncoTreat test performed based on tumor type and pathology with sufficient cellularity/DNA/RNA; as testing will not be able to be performed on those whose tumor tissue biopsies contain less than 50% of tumor. We had added the latter statement to clarify.

b. (lines 159-165) can you make it clearer who is doing the biopsy and that Columbia University Medical Center is doing the Darwin testing. I think this is clearer in Fig 3, than it is in the text.

Response: Thank you for pointing this out. We have added a clarifying statement.

c. Are you going to report how many patients were ineligible for patient selection overall and why (e.g. no consent, NEAT score)? This would also be useful to know for patient acceptability – e.g. out of how many patients >18 with up to 10 lesions and classify as oligometastases, were included in the trial.

Response: If there are patients who initially consent to the trial but do not ultimately receive Darwin testing results, the reason for attrition will be reported (eg, voluntary withdrawal, insufficient tumor, death, etc.). Additionally, identifying any unknown barriers to testing is one of our feasibility endpoints. The overall potential pool of patients is those with metastatic cancer referred to radiation oncology, but most of these patients don’t qualify because they have >10 lesions, a NEAT score that’s too low, or they haven’t failed first-line systemic therapy. We’re not including these ineligible patients because they haven’t signed a consent form and therefore are not formally part of the study.

5. Would the long term goal be for an RCT to evaluate the efficacy of precision medicine using the Darwin tests for metastatic cancer survival? If so, would it be useful to include the acceptability of randomisation to precision medicine, in theory, for both patients and clinicians?

Response: Thank you for your comment. The long-term goal, with success of this feasibility trial is an RCT. We have added a reflection on this to our Discussion section.

REVIEWER #2: Very Minor Thought

This appears to be a well-designed feasibility trial with the feasibility objectives to be correctly summarized without formal statistical analysis. As the authors will no doubt concede, statistical significance is unlikely when the suggested comparisons of progression free survival and overall survival are made. However, 95%CI intervals for the HRs estimated would be informative.

Response: That you so much for this added value. We have added 95% CI for HR to our statistical analysis section.

7. PLOS authors have the option to publish the peer review history of their article (what does this mean?). If published, this will include your full peer review and any attached files.

Do you want your identity to be public for this peer review? For information about this choice, including consent withdrawal, please see our Privacy Policy.

Reviewer #1: No

Reviewer #2: No

Response: Thank you for your review.

---

## [Decision Letter · Decision Letter 1]

21 May 2025

Feasibility trial of Darwin OncoTreat and OncoTarget precision medicine testing to improve outcomes for patients with limited metastatic disease that failed first-line systemic therapy

PONE-D-24-45282R1

Dear Dr. Radigan,

We’re pleased to inform you that your manuscript has been judged scientifically suitable for publication and will be formally accepted for publication once it meets all outstanding technical requirements.

Kind regards,

Andres Mauricio Acevedo-Melo, M.D.

Academic Editor

PLOS ONE

Additional Editor Comments (optional):

All comments have been succesfully addressed

Reviewers' comments:

Reviewer's Responses to Questions

**Comments to the Author**

1. Does the manuscript provide a valid rationale for the proposed study, with clearly identified and justified research questions?

Reviewer #2: Yes

2. Is the protocol technically sound and planned in a manner that will lead to a meaningful outcome and allow testing the stated hypotheses?

Reviewer #2: Yes

3. Is the methodology feasible and described in sufficient detail to allow the work to be replicable?

Reviewer #2: Yes

4. Have the authors described where all data underlying the findings will be made available when the study is complete?

Reviewer #2: Yes

5. Is the manuscript presented in an intelligible fashion and written in standard English?

Reviewer #2: Yes

You may also provide optional suggestions and comments to authors that they might find helpful in planning their study.

Reviewer #2: As I suggested, the authors have now included 95%CI for the HRs concerned.

I have no further comments to add.

I wish them well with their study

**Do you want your identity to be public for this peer review?** For information about this choice, including consent withdrawal, please see our Privacy Policy

Reviewer #2: No

---

## [Editor Report · Acceptance letter]

PONE-D-24-45282R1

PLOS ONE

Dear Dr. Radigan,

I'm pleased to inform you that your manuscript has been deemed suitable for publication in PLOS ONE. Congratulations! Your manuscript is now being handed over to our production team.

Kind regards,

on behalf of

Dr. Andres Mauricio Acevedo-Melo

Academic Editor

PLOS ONE